# Interpreting the Response of Forest Stock Volume with Dual Polarization SAR Images in Boreal Coniferous Planted Forest in the Non-Growing Season

**Huanna Zheng** [1,2,3], **Jiangping Long** [1,2,3], **Zhuo Zang** [1,2,3,*], **Hui Lin** [1,2,3], **Zhaohua Liu** [1,2,3], **Tingchen Zhang** [1,2,3] and **Peisong Yang** [1,2,3]

1 Research Center of Forestry Remote Sensing & Information Engineering, Central South University of Forestry & Technology, Changsha 410004, China; 20211100037@csuft.edu.cn (H.Z.); longjiangping@csuft.edu.cn (J.L.); t19911090@csuft.edu.cn (H.L.); 20200100005@csuft.edu.cn (Z.L.); 20220100015@csuft.edu.cn (T.Z.); 20210100004@csuft.edu.cn (P.Y.)
2 Hunan Provincial Key Laboratory of Forestry Remote Sensing Based Big Data & Ecological Security, Changsha 410004, China
3 Key Laboratory of National Forestry and Grassland Administration on Forest Resources Management and Monitoring in Southern China, Changsha 410004, China
* Correspondence: zangzhuo@csuft.edu.cn; Tel.: +86-138-7310-2351

**Abstract:** Polarimetric Synthetic Aperture Radar (PolSAR) images with dual polarization modes have great potential to map forest stock volume (FSV) by excellent penetration capabilities and distinct microwave scattering processes. However, the response of these SAR data to FSV is still uncertain in the non-growing season. To further interpret the response of FSV to different dual polarization SAR images, three types of dual polarization SAR images (GF-3, Sentinel-1, and ALOS-2) were initially acquired in coniferous planted forest in the non-growing season. Then, sensitivity between FSV and all alternative features extracted from each type of SAR image was analyzed to express the response of FSV to dual polarization SAR images with bands and polarization modes in the non-growing season in deciduous (Larch) and evergreen (Chinese pine) forests. Finally, mapped FSV using single and combined dual polarization images were derived by optimal feature sets and four machine learning models, respectively. The combined effects were also analyzed to clarify the difference of bands and polarization modes in deciduous and evergreen forests in the non-growing season. The results demonstrated that the backscattering energy from different sensors is significantly different in Chinese pine, and the difference is gradually reduced in Larch forests. It is also implied that the polarization mode is more important than penetration capability in mapping forest FSV in deciduous forest in the non-growing season. By comparing the accuracy of mapped FSV using single and combined images, combined images have more capability to improve the accuracy and reliability of mapped FSV. Meanwhile, it is confirmed that compensation effects with bands and polarization modes not only have great potential to delay the saturation phenomenon, but also have the capability to reduce errors caused by overestimation.

**Keywords:** forest stock volume (FSV); dual polarization SAR; polarization modes; non-growing season; coniferous planted forest

## 1. Introduction

Planted forests play a significant role in making substantial contributions to climate change mitigation and promoting the carbon cycle within ecosystems. The forest stock volume (FSV) is regarded as one of key indices in evaluating the quality of planted forests [1,2]. Normally, traditional methods of investigating FSV are obtained by time-consuming and costly ground measurements, with which it is hard to satisfy the requirements of modern forest resource monitoring [3]. In the last twenty years, remote sensing technology has provided a more promising approach to indirectly mapping FSV with a few ground-measured

samples for large regions [4]. However, electromagnetic signals reflected from forests are often prevented by cloud cover using optical remote sensing images [5–7]. Furthermore, optical remote sensing images are still useless for forests during the deciduous season. To overcome these disadvantages, Polarimetric Synthetic Aperture Radar (PolSAR) images have a great potential to map FSV by excellent penetration capabilities and distinct microwave scattering processes [8].

Commonly, various bands (X, C, and L-bands), and various polarization modes (single, dual, and quad) SAR images have been widely applied in mapping forest parameters [9–11]. Previous results have demonstrated that polarization features extracted from quad polarization SAR images had a remarkable sensitivity to FSV [12]. However, the number of images acquired with the dual polarization mode is much larger than that with the quad polarization mode among these available SAR sensors [9]. Therefore, dual polarization SAR images are still a main option to map forest parameters in large regions. Furthermore, the sensitivity of features related to FSV is affected by the ability to penetrate the forest. As the wavelength increases, the ability of the SAR signal to penetrate the forest increases [13,14]. Previous studies have indicated that SAR images scattering of different bands reflects the FSV in different parts of the forest, with C-band SAR images reflecting more forest canopy information, while the L-band penetrates the forest canopy and captures vertical information. Most previous studies have shown that L-band SAR images showed greater sensitivity to FSV than C and X-band SAR images in the growing season [15]. However, most of these studies have focused on evergreen deciduous and coniferous forests during the growing season. Therefore, for evergreen and deciduous forests in the non-growing season, it is valuable to further interpret the response of FSV with dual polarization SAR images with various bands, such as C and L-band SAR images.

Normally, features extracted from dual polarization SAR images with different bands show different sensitivities to FSV. Recently, two polarization combinations have been widely provided for dual polarization images, such as HH (horizontal transmission and horizontal reception) and HV polarization (horizontal transmission and vertical reception) for GF-3 and ALOS-2, VV (vertical transmission and vertical reception) and VH (vertical transmission and horizontal reception) for Sentinel-1 [16]. Previous results have shown that the sensitivity between the backscatter coefficient of cross-polarization (HV and VH) and FSV is significantly higher than co-polarization (HH and VV) [17–19]. Because of the different penetration abilities, some studies also showed that C-band SAR images were more closely related to forest crowns, while L-band images received more reflection information of trunks and branches [20–23]. Although L-band SAR images have higher saturation levels for mapping forest FSV, they still affect the accuracy of estimating forest FSV, especially for mature planted forests [24–27]. Furthermore, previous results have shown that the accuracy of mapping FSV had been greatly improved by using multiple bands SAR images [28–30]. However, it is still an uncertain question to clarify the combined effects of bands and polarization modes for different tree species and their growth conditions.

Additionally, forest types are also one of the key factors that influences the sensitivity between forest FSVs and features extracted from PolSAR images [31,32]. Previous studies have indicated that the accuracy and saturation levels of mapping FSV have significant differences in temperate conifers and boreal conifers forests using the same bands of SAR images [33]. Moreover, the accuracy of mapped AGB in deciduous and evergreen forests has great gaps using Sentinel-1 images in the same region [28]. Therefore, it is valuable to analyze seasonal factors for mapping forest FSV using dual polarization SAR images in deciduous forests.

The objective is to clarify the response of forest FSV to various dual polarization SAR images with different bands and polarization modes in deciduous (Larch) and evergreen (Chinese pine) forests and interpret the combined effects of mapping forest FSV using GF-3, Sentinel-1, and ALOS-2 dual SAR images acquired in the non-growing season. In this study, several types of alternative features were initially extracted from three types of acquired dual polarization SAR images, and the sensitivity between features and forest FSV was

analyzed to clarify the response of forest FSV to bands and polarization modes. Finally, using the sequential forward selection method and four machine learning models, mapped FSVs were derived from various combined images with bands and polarization modes in evergreen and deciduous forests in the non-growing season.

## 2. Materials and Methods

### 2.1. Studying Area

The study was conducted in the area of Wangyedian Experimental Forest Farm, located in Harqin Banner, Inner Mongolia Autonomous Region, China (Figure 1). The geographic coordinates of the forest farm are east longitude ranged from 118°09′ to 118°30′ and north latitude ranged from 41°21′ to 41°39′, respectively. The terrain is characterized by mid-low mountainous areas, with elevations ranging from 600 m to 1890 m. According to statistics data from 2016, the area of forest cover is up to 23,300 hectares, with a forest coverage rate of 93%. The total volume of timber is nearly 1.527 million m$^3$, and 49.78% of the research area (approximately 11,600 hectares) is covered by planted forests, mainly tree species of Larch (***Larix principles-Ruprecht and Larixolgensis***) and Chinese pine (***Pinus tabuliformis***).

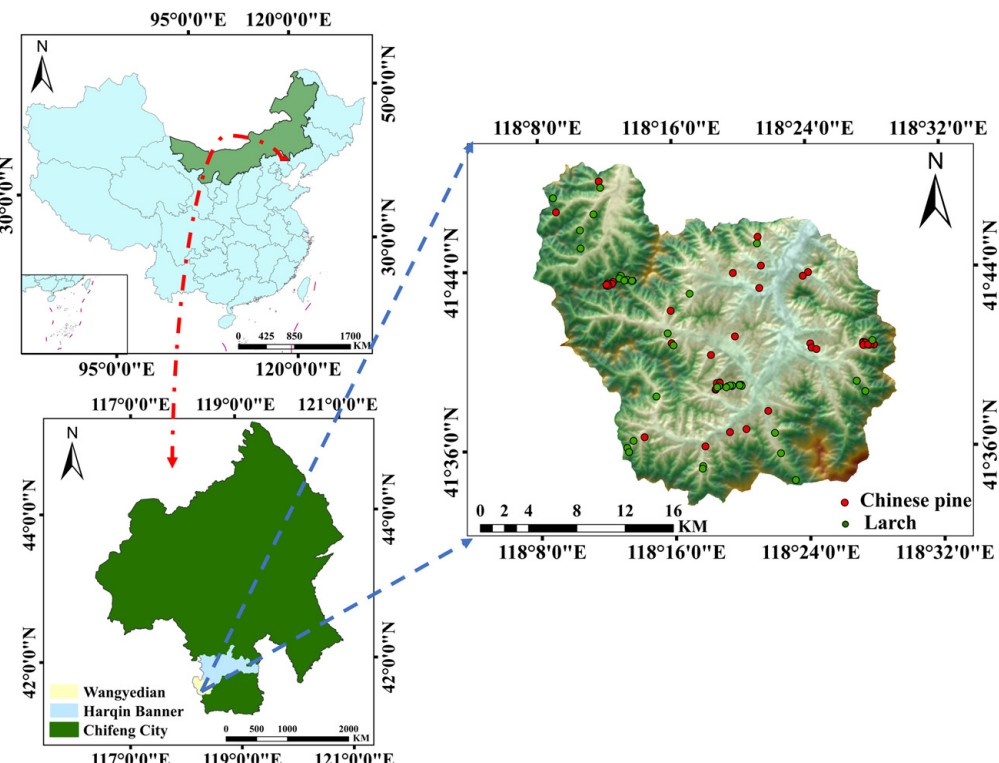

**Figure 1.** The location of the study area and maps of ground samples.

### 2.2. Ground Data

For mapping forest FSV, the field-measured data in this study were collected in October 2017. Based on the age and tree species, a total of 81 ground samples (Larch: 38 and Chinese pine: 43) were determined by a stratified random sampling method. For each sample with a size of 25 m × 25 m, the position of corners and centers were precisely surveyed by GNSS with errors of less than 10 cm. And then, the tree height, DBH (diameter at breast height), and crown size of each tree in sample were measured. The volume of each tree was calculated using the bivariate volume equation based on the collected tree parameters (Table 1). Finally, FSV was obtained by the sum of all trees in one sample. The distribution of FSV is illustrated in Figure 2.

**Table 1.** The bivariate volume formulas of two planted tree species.

| Tree Species | Volume Formula | Note |
|---|---|---|
| Chinese pine | $V = 0.013464 - 0.001967 \times D + D^2 + 0.000628 \times D \times H$ $+ 0.000032 \times H \times D^2 - 0.003173 \times H$ | V: Volume D: Diameter |
| Larch | $V = -0.001498 + 0.00007 \times D^2 + 0.000901 \times H + 0.000032 \times H \times D^2$ | H: Height |

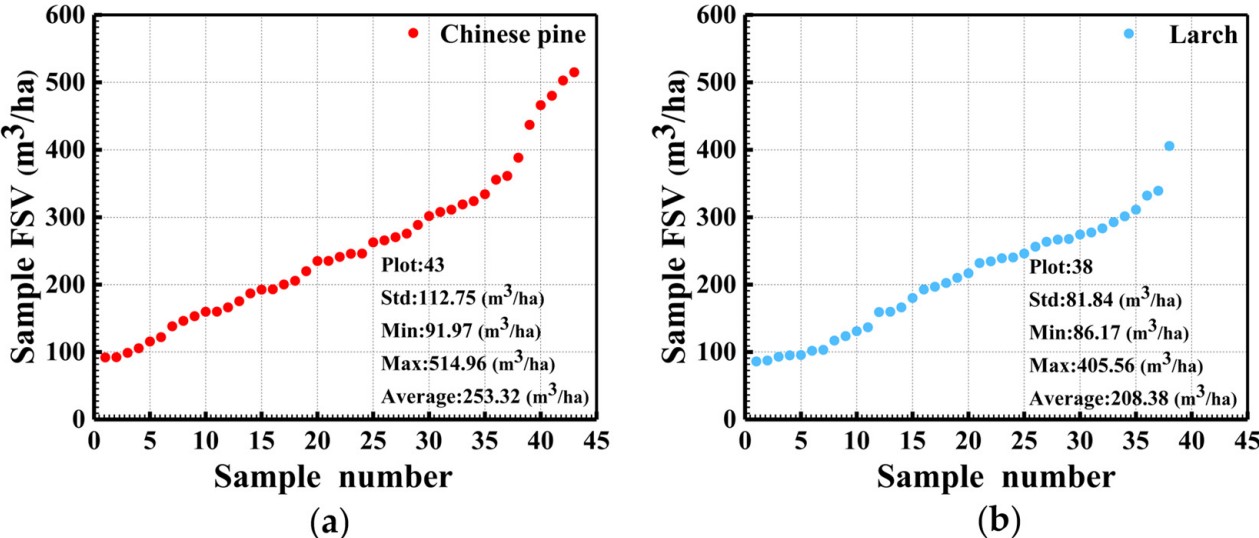

**Figure 2.** The distribution of sorted FSV; (**a**) is for Chinese pine and (**b**) is for Larch.

*2.3. Dual Polarization SAR Images and DEM*

In this study, forest FSV was estimated using three types of C and L-bands dual polarimetric SAR images acquired in March 2017, including GF-3 (C-band), Sentinel-1 (C-band), and ALOS-2 (L-band). Table 2 listed the information of acquired multi-band dual polarization SAR images. For GF-3 and ALOS-2 images, the polarization mode is HH and HV, and for Sentinel-1 images, the polarization mode is VV and VH. Additionally, an open-sourced digital elevation model (DEM) with a spatial resolution of 30 m was also employed to geocode the SAR images.

**Table 2.** The lists of acquired multi-band dual polarization SAR images.

| Number | Band | Acquired Time | Sensors | Polarization Modes | Incidence Angle | Resolution |
|---|---|---|---|---|---|---|
| 1 | C-band | 2017.03.18 | GF-3 | HH and HV | 38.57° | 2.25 m × 3.12 m |
| 2 | C-band | 2017.03.04 | Sentinel-1 | VV and VH | 39.50° | 2.32 m × 13.89 m |
| 3 | L-band | 2017.03.09 | ALOS-2 | HH and HV | 31.41° | 4.29 m × 3.09 m |

*2.4. Dual Polarization SAR Image Pre-Processing*

Normally, several image pre-processing steps were employed before extracting alternative features from dual polarization SAR images. Firstly, radiation calibration was applied to convert the amplitude information into quantitative values. Secondly, the multi-look processing and adaptive Frost filtering with a window size of 5 × 5 were performed to smooth the coherent speckle noise of the SAR image. In addition, topographic radiation correction was also needed to reduce the topographic influences, such as shrinkage, overlay, and shadowing. Finally, all images were geocoded using external digital elevation model (DEM) with spatial resolution of 30 m. All of the above pre-processing steps were done in ENVI 5.6 software.

## 2.5. Feature Extraction

For dual polarization SAR images, it is rather necessary to extract enough features for mapping forest FSV, and backscattering coefficients with different polarizations are initially extracted. Commonly, these backscattering coefficients of dual polarization SAR include σHH, σHV (GF-3 and ALOS-2) and σVH, σVV(Sentinel-1), for a total of six backscattering coefficients. For increasing the number of alternative features, mathematical operations between backscattering coefficients with different polarizations were applied and eighteen derived features [1] were extracted from each type of SAR image, for a total of 54 derived features from three SAR images in this study (Table 3).

**Table 3.** The list of backscattering coefficients and their derived features extracted from three acquired SAR images.

| Number | Feature | Note | Number | Feature | Note |
|---|---|---|---|---|---|
| 1 | HH (geo), VV (geo) | $\sqrt{a^2 + b^2}$ | 12 | X8 | $\sigma HV/(\sigma HH + \sigma HV)$, $\sigma VH/(\sigma VV + \sigma VH)$ |
| 2 | HV (geo), VH (geo) | $\sqrt{a^2 + b^2}$ | 13 | X9 | $(\sigma HH)^2$, $(\sigma VV)^2$ |
| 3 | σHH, σVV | dB | 14 | X10 | $(\sigma HV)^2$, $(\sigma VH)^2$ |
| 4 | σHV, σVH | dB | 15 | X11 | $(X1)^2$ |
| 5 | X1 | σHH + σHV, σVV + σVH | 16 | X12 | $(X2)^2$ |
| 6 | X2 | σHH-σHV, σVV-σVH | 17 | X13 | $(X3)^2$ |
| 7 | X3 | σHH/σHV, σVV/σVH | 18 | X14 | $(X4)^2$ |
| 8 | X4 | σHV/σHH, σVH/σVV | 19 | X15 | $(X5)^2$ |
| 9 | X5 | σHH*σHV, σVV × σVH | 20 | X16 | $(X6)^2$ |
| 10 | X6 | (σHH-σHV)/(σHH + σHV), (σVV-σVH)/(σVV + σVH) | 21 | X17 | $(X7)^2$ |
| 11 | X7 | σHH/(σHH + σHV), σVV/(σVV + σVH) | 22 | X18 | $(X8)^2$ |

Note: a and b are the real and imaginary components of SLC image; HH (geo), VV (geo), HV (geo) and VH (geo) are the intensity of HH, VV, HV and VH polarization with a linear form; σHH, σVV, σHV and σVH are backscattering coefficients of HH, VV, HV, and VH polarization; X1–X18 are derived features related to backscattering coefficients.

Furthermore, Gray Level Co-occurrence Matrix (GLCM) was also employed to extract textural information from each polarized intensity image (σHH, σHV, σVH, and σVV); there were eight textural features, including mean, variance, homogeneity, contrast, dissimilarity, entropy, second moment, and correlation, which were obtained from each image with various window sizes ($5 \times 5$, $7 \times 7$, $9 \times 9$). In this study, 144 texture features were extracted from three SAR images.

## 2.6. Feature Selection and Models

To obtain the optimal feature set, the sensitivity between features and FSV was firstly evaluated by the Pearson correlation coefficient and the absolute values of the correlation were also obtained. Then the features were ordered in descending order according to the Pearson correlation coefficient. After obtaining sorted features, the sequential forward selection method and four machine learning models were employed to construct wrapped feature selection methods [1,34]. Finally, optimal feature sets were obtaining by removing the features with poor contributions to the accuracy of estimating FSV. In this study, K-Nearest Neighbors Regression (KNN), Multiple Linear Regression (MLR), Random Forest Regression (RF), and Support Vector Machine Regression (SVR) were employed to map FSV using wrapped feature selection methods. All four regression models were constructed in R software, where the parameters to be tuned for KNN are the number of neighbours K, the parameters to be tuned for RF are the number of random trees (ntree) and the number of randomly chosen variables to split each node in the tree (mtry), the parameters to be tuned for SVM are the penalty coefficient (C) and gamma, and no parameters are needed to be tuned for MLR. Furthermore, combined images of different strategies (ALOS-2 + GF-3,

ALOS-2 + Sentinel-1, Sentinel-1 + GF-3, and ALOS-2 + Sentinel-1 + GF-3) were formed to interpret the combined effects of mapping forest FSV. The framework of mapping FSV with combined dual polarization SAR data is illustrated in Figure 3. R 4.2.3 software was used to select features and construct four machine learning models.

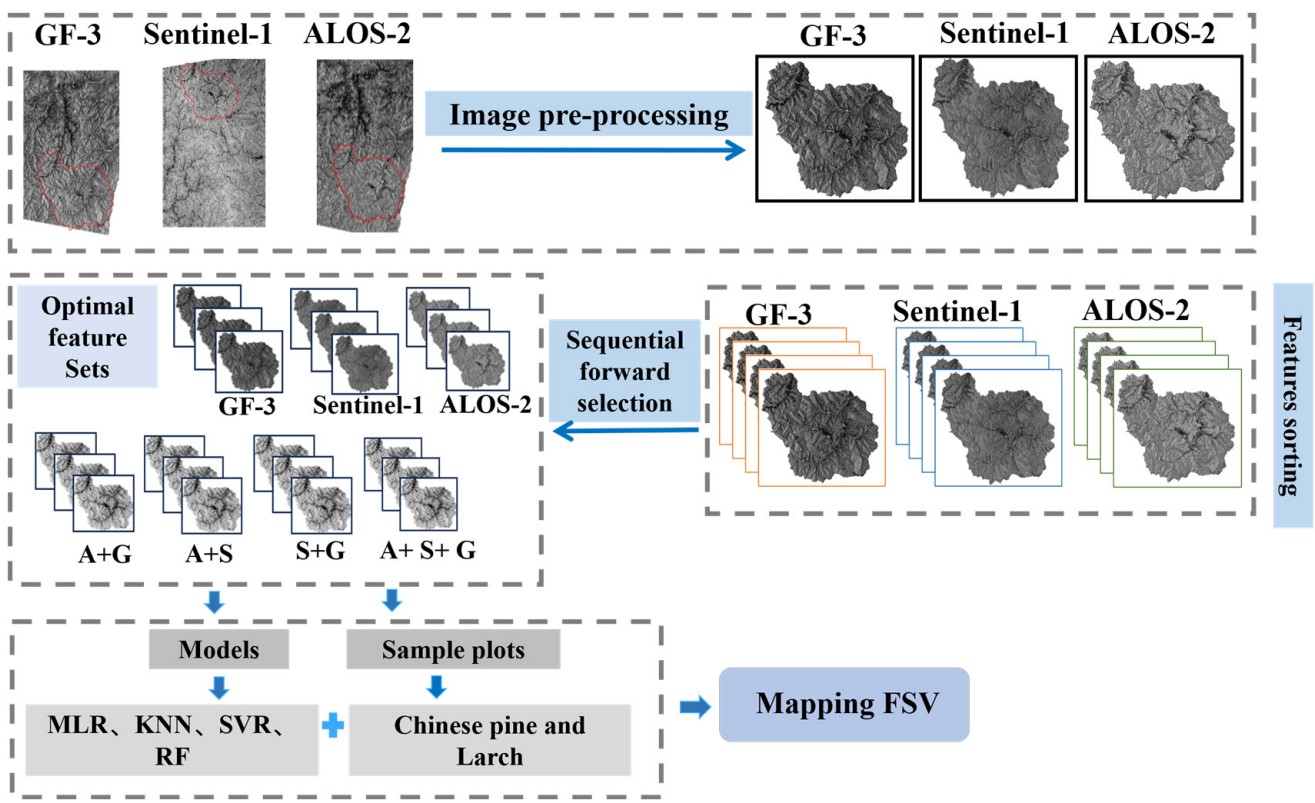

**Figure 3.** Framework for mapping FSV with dual polarization SAR data. (A + G:ALOS-2 + GF-3; A + S:ALOS-2 + Sentinel-1; A + S + G:ALOS-2 + Sentinel-1 + GF-3).

### 2.7. Accuracy Evaluation

Based on the number of samples, LOOCV was employed to evaluate the accuracy of mapping FSV among various combined SAR images. Specifically, Root Mean Square Error (RMSE), Relative Root Mean Square Error (rRMSE), and Coefficient of Determination (R-squared, $R^2$) were selected as the accuracy evaluation metrics for evaluating models. The formulas of these evaluation metrics are as follows.

$$\text{RMSE} = \sqrt{\frac{1}{N}\sum_{i=1}^{n}(\hat{y}_i - y_i)} \tag{1}$$

$$\text{rRMSE} = \left(\frac{\text{RMSE}}{y_i}\right) \times 100\% \tag{2}$$

$$R^2 = 1 - \frac{\sum_{i=1}^{n}(\hat{y}_i - y_i)^2}{\sum_{i=1}^{n}(y_i - y)^2} \tag{3}$$

where $\hat{y}_i$ indicates the predicted FSV, $y_i$ is the measured FSV, y indicates the mean of the measured FSV, and n indicates the total number of samples.

## 3. Results

### 3.1. The Sensitivity between Features and Forest FSV

For planted Chinese pine and Larch in the non-growing season, the backscattering energy with different bands and polarization modes was extracted to explore the response

of FSV (Figure 4). The results illustrated that the scattering energy from the HH and VV polarization mode was obviously greater than that from HV and VH, because of differences in scattering mechanisms. Moreover, the difference of backscattering energy from different sensors is significant in planted Chinese pine forest (Figure 4a,b), and the difference among these bands and polarization modes is gradually reduced in planted Larch forests (Figure 4c,d). Specially, it was difficult to distinguish the difference of backscattering energy between ALOS-2 and Sentinel-1 in co-polarization and cross-polarization modes in planted Larch forests. For the same polarization mode, differences of penetration capacity with L and C-bands can still be clearly identified in deciduous forest in the non-growing season. It is implied that the difference of backscattering energy in bands and polarization modes can be recognized in evergreen and deciduous forest in the non-growing season.

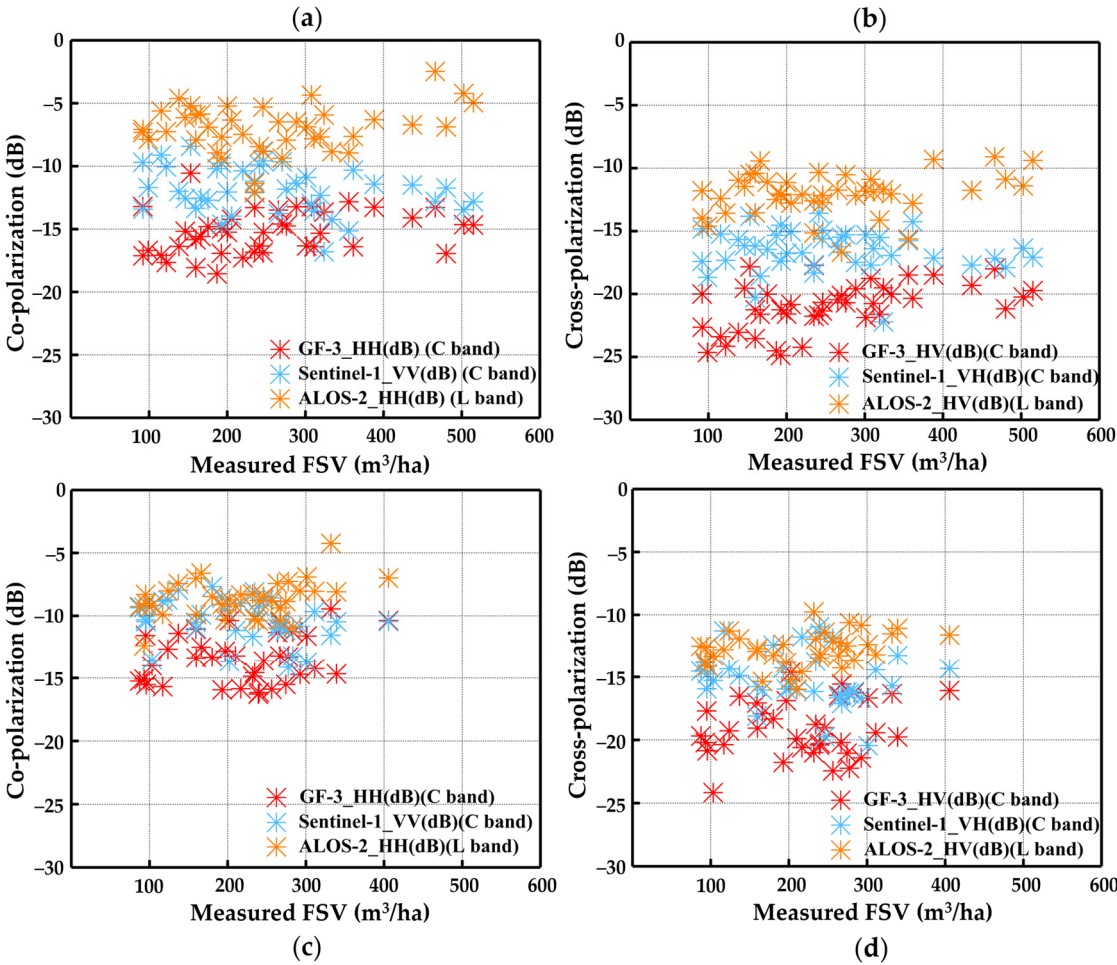

**Figure 4.** The scatterplot of backscattering energy with multi-bands and polarization modes; (**a**,**b**) are for planted Chinese pine, (**c**,**d**) are for planted Larch.

Moreover, the absolute values of the Pearson correlation coefficient between FSV and all alternative features extracted from three types of sensors were used to further explore the sensitivity. The top ten sensitivity features of planted Chinese pine and Larch in different sensors were illustrated in Figure 5, respectively. For planted Chinese pine, the absolute values of the Pearson correlation coefficient ranged from 0.43 to 0.51 for GF-3, from 0.47 to 0.59 for Sentinel-1, and from 0.29 to 0.46 for ALOS-2, respectively (Figure 5). It was found that the texture features from three types of dual polarization SAR images correlated well with the FSV, and the absolute values of Pearson correlation between FSV and texture features extracted from cross-polarization images had higher sensitivity than those extracted from co-polarization images (Figure 5). For planted Larch in the non-growing season, the texture features from three types of dual polarization SAR images

also correlated well with the FSV (Figure 5), and the absolute values of Pearson correlation between FSV and texture features extracted from ALOS-2 were obviously higher than those extracted from GF-3 and Sentinel-1 (Figure 5). And in the optimal feature set, some texture features were selected as important features by the sequential forward selection method. It is demonstrated that features extracted from three types of dual polarization SAR images have great potential to map FSV in evergreen and deciduous forest in the non-growing season.

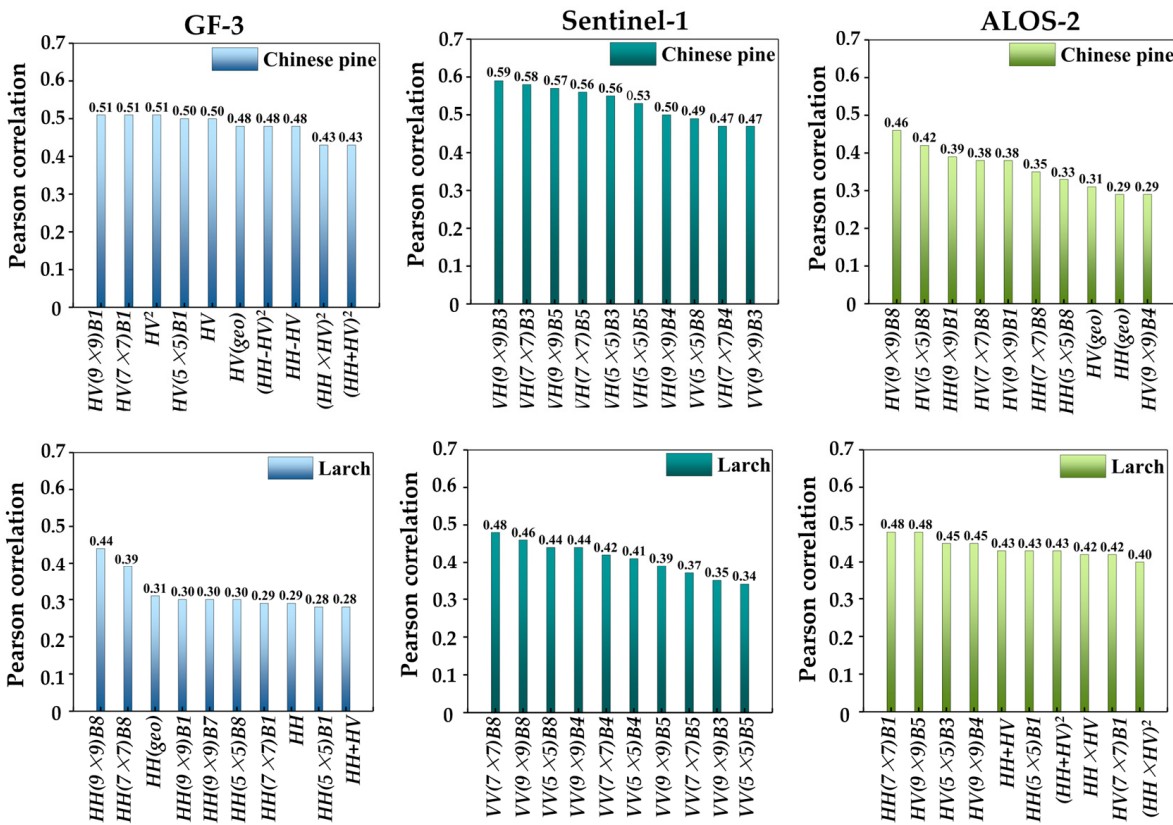

**Figure 5.** The features with sensitivity ranking within the top 10 of planted Chinese pine and Larch in different sensors.

### 3.2. The Results of Estimated FSV Using Single Dual Polarization SAR Images

To further evaluate the potential of these dual polarization SAR data in mapping FSV in the non-growing season, the sequential forward selection and four machine learning models (MLR, KNN, SVR, and RF) were employed to construct wrapped feature selection methods, and the optimal feature sets were ultimately obtained for each type of dual polarization SAR images. The results of estimated FSV of Chinese pine and Larch are shown in Table 4. Among the three types of dual polarization SAR data, the differences of rRMSE in different models were less than five percentage points.

For Chinese pine, the values of rRMSE using optimal feature sets ranged from 30.58% to 34.08% for GF-3, from 31.60% to 37.99% for Sentinel-1, and from 29.33% to 33.07% for ALSO-2, respectively (Table 4). The highest average accuracy of mapping FSV was obtained using dual polarization ALSO-2 images. The results indicated that the L-band dual polarization images (ALSO-2) have more potential than C-band images (GF-3 and Sentinel-1) in mapping forest FSV in evergreen forest. For planted Larch, the values of rRMSE using optimal feature sets ranged from 27.34% to 32.80% for GF-3, from 24.47% to 28.49% for Sentinel-1, and from 30.05% to 33.13% for ALSO-2, respectively (Table 4). The highest average accuracy of mapping FSV was obtained using dual polarization Sentinel-1 images. The results also showed that the accuracy of mapping FSV using C-band dual

polarization SAR images (GF-3 and Sentinel-1) was slightly higher than using L-band dual polarization SAR images (ALSO-2). It is demonstrated that polarization mode is more important than penetration capability of bands in mapping forest FSV in deciduous forest in the non-growing season.

**Table 4.** The table of accuracy indices in mapping FSV using each type of dual polarization SAR images.

| Data | Models | Chinese Pine | | | | | Larch | | | | |
|---|---|---|---|---|---|---|---|---|---|---|---|
| | | RMSE (m³/ha) | rRMSE (%) | R² | Features Number | Average rRMSE (%) | RMSE (m³/ha) | rRMSE (%) | R² | Features Number | Average rRMSE (%) |
| GF-3 | MLR | 87.48 | 34.08 | 0.40 | 6 | 32.61 | 62.92 | 28.34 | 0.35 | 4 | 30.00 |
| | KNN | 85.50 | 33.31 | 0.43 | 19 | | 69.95 | 31.51 | 0.20 | 7 | |
| | SVR | 78.51 | 30.58 | 0.51 | 4 | | 60.68 | 27.34 | 0.40 | 4 | |
| | RF | 83.36 | 32.48 | 0.46 | 12 | | 72.79 | 32.80 | 0.14 | 13 | |
| Sentinel-1 | MLR | 97.51 | 37.99 | 0.27 | 5 | 35.23 | 58.40 | 26.31 | 0.44 | 4 | 26.63 |
| | KNN | 88.98 | 34.66 | 0.38 | 15 | | 60.45 | 27.23 | 0.41 | 12 | |
| | SVR | 81.11 | 31.60 | 0.49 | 5 | | 54.31 | 24.47 | 0.52 | 15 | |
| | RF | 94.10 | 36.66 | 0.31 | 11 | | 63.24 | 28.49 | 0.35 | 13 | |
| ALOS-2 | MLR | 77.24 | 30.09 | 0.53 | 5 | 31.21 | 67.64 | 30.47 | 0.25 | 2 | 31.06 |
| | KNN | 83.03 | 32.34 | 0.46 | 8 | | 66.70 | 30.05 | 0.28 | 12 | |
| | SVR | 75.29 | 29.33 | 0.56 | 7 | | 67.68 | 30.58 | 0.25 | 6 | |
| | RF | 84.89 | 33.07 | 0.44 | 8 | | 73.53 | 33.13 | 0.12 | 13 | |

To further validate the performance of three SAR data in mapping FSV, the scatter plots and residuals between predicted and measured FSV of samples were demonstrated using the models with the highest accuracy of results from each type of data (Figure 6). The results demonstrated that the accuracy of mapping FSV in Chinese pine forests was higher than that in Larch forests in the non-growing season. Furthermore, the number of underestimation samples in Larch forests was obviously larger than in Chinese Pine forests. To improve the accuracy of mapping FSV, it is valuable to further analyze the response of FSV with a combined strategy with bands and polarization modes.

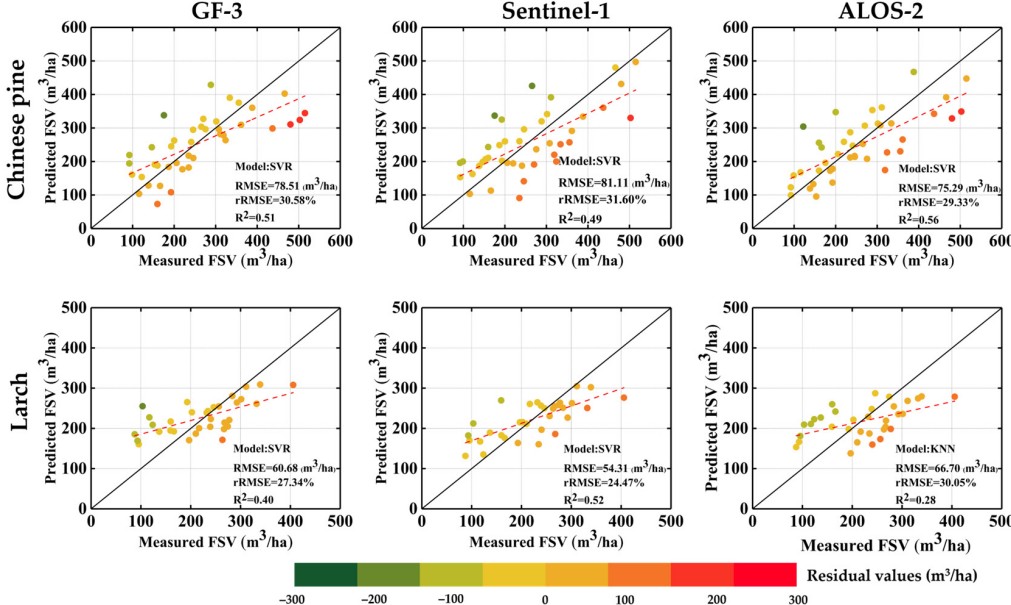

**Figure 6.** The scatter plots of estimating FSV using the models with the highest accuracy of results from each type of data; the red dashed line is the fitted line, and the color of the points is determined by the residual between the predicted and ground-measured FSV.

### 3.3. The Results of Estimated FSV Using Combined Images of Different Strategies

To further analyze the potential of combined dual polarization SAR data with different bands and polarization modes, the combined images of different strategies were used to map the FSV in Chinese pine and Larch forests in the non-growing season, respectively. The optimal feature sets of combined images were also obtained by the forward feature selection method and four machine learning models (MLR, KNN, SVR, and RF). The accuracy indices of mapping FSV using combined images of different strategies were listed in Table 5.

**Table 5.** The accuracy indices of mapping FSV using combined images of different strategies.

| Combined Images | Model | Chinese Pine | | | | | Larch | | | | |
|---|---|---|---|---|---|---|---|---|---|---|---|
| | | RMSE (m³/ha) | rRMSE (%) | R² | Selected Features | Average rRMSE (%) | RMSE (m³/ha) | rRMSE | R² | Selected Features | Average rRMSE (%) |
| ALOS-2 + GF-3 | MLR | 71.40 | 27.81 | 0.60 | 9 | 28.93 | 60.54 | 27.27 | 0.40 | 4 | 27.68 |
| | KNN | 72.21 | 28.13 | 0.59 | 21 | | 61.36 | 27.65 | 0.39 | 13 | |
| | SVR | 72.66 | 28.31 | 0.58 | 8 | | 55.20 | 24.87 | 0.50 | 9 | |
| | RF | 80.45 | 31.46 | 0.49 | 21 | | 68.68 | 30.94 | 0.23 | 13 | |
| ALOS-2 + Sentinel-1 | MLR | 67.64 | 26.35 | 0.64 | 4 | 29.56 | 61.69 | 27.80 | 0.38 | 5 | 26.51 |
| | KNN | 74.35 | 28.96 | 0.57 | 22 | | 57.60 | 25.95 | 0.46 | 12 | |
| | SVR | 75.83 | 29.54 | 0.55 | 11 | | 60.09 | 27.07 | 0.41 | 14 | |
| | RF | 85.67 | 33.37 | 0.43 | 15 | | 55.95 | 25.21 | 0.49 | 13 | |
| Sentinel-1 + GF-3 | MLR | 71.06 | 27.68 | 0.61 | 3 | 28.22 | 65.29 | 29.41 | 0.31 | 4 | 26.92 |
| | KNN | 64.81 | 25.25 | 0.67 | 25 | | 55.93 | 25.20 | 0.49 | 20 | |
| | SVR | 67.67 | 26.36 | 0.64 | 11 | | 54.35 | 24.49 | 0.52 | 11 | |
| | RF | 86.17 | 33.57 | 0.42 | 20 | | 63.42 | 28.57 | 0.34 | 15 | |
| ALOS-2 + Sentinel-1 + GF-3 | MLR | 70.02 | 27.28 | 0.62 | 5 | 25.60 | 61.83 | 27.86 | 0.38 | 4 | 26.18 |
| | KNN | 57.59 | 22.43 | 0.74 | 21 | | 53.55 | 24.13 | 0.53 | 16 | |
| | SVR | 54.67 | 21.30 | 0.77 | 9 | | 56.91 | 25.64 | 0.47 | 5 | |
| | RF | 80.59 | 31.40 | 0.49 | 18 | | 60.15 | 27.10 | 0.41 | 13 | |

Using combined images of different strategies, the accuracy of mapped FSV was obviously improved in Chinese pine and Larch forests. The values of $R^2$ ranged from 0.42 to 0.77 for Chinese pine forest and from 0.23 to 0.53 for Larch forests, respectively (Table 5). Moreover, the values of average rRMSE obtained from combined dual polarization images (ranged from 25.60% to 29.56%) were significantly smaller than those from single dual polarization images (ranged from 31.21% to 35.23%) in Chinese pine forests (Table 5), and the best results were obtained from a combination of three types of sensors (ALOS-2 + Sentinel-1 + GF-3). For Larch forests, the values of average rRMSE obtained from combined dual polarization images (ranged from 26.18% to 27.68%) were slightly smaller than those from single dual polarization images (ranged from 26.63% to 31.06%) (Table 5), and the best results were also obtained from a combination of three types of sensors (ALOS-2 + Sentinel-1 + GF-3). The results demonstrated that combined images with bands and polarization modes have the capability to improve the accuracy and reliability of mapped FSV, and the degree of accuracy improvement is related to tree species and season.

To further interpret the response of FSV with different types of dual polarization SAR images, scatter plots and residuals between predicted and measured FSV were plotted using the best results of each type of combination (Figure 7). The results demonstrated that combined dual polarization images with different strategies can improve accuracy due to a significant improvement in underestimation. It is inferred that the compensation effect with bands and polarization modes has exciting potential to delay the saturation phenomenon. Additionally, the results also demonstrated that overestimated results which occurred in low FSV were also obviously improved using combined dual polarization SAR images with different strategies, and the degree of improvement in planted Chinese pine was higher than that in planted Larch. In this study, the maps of forest FSV were finally generated by the best models using a combination of three types of dual polarization SAR images (ALOS-2 + Sentinel-1 + GF-3) in evergreen (Figure 8a) and deciduous forest (Figure 8b) in the non-growing season, respectively.

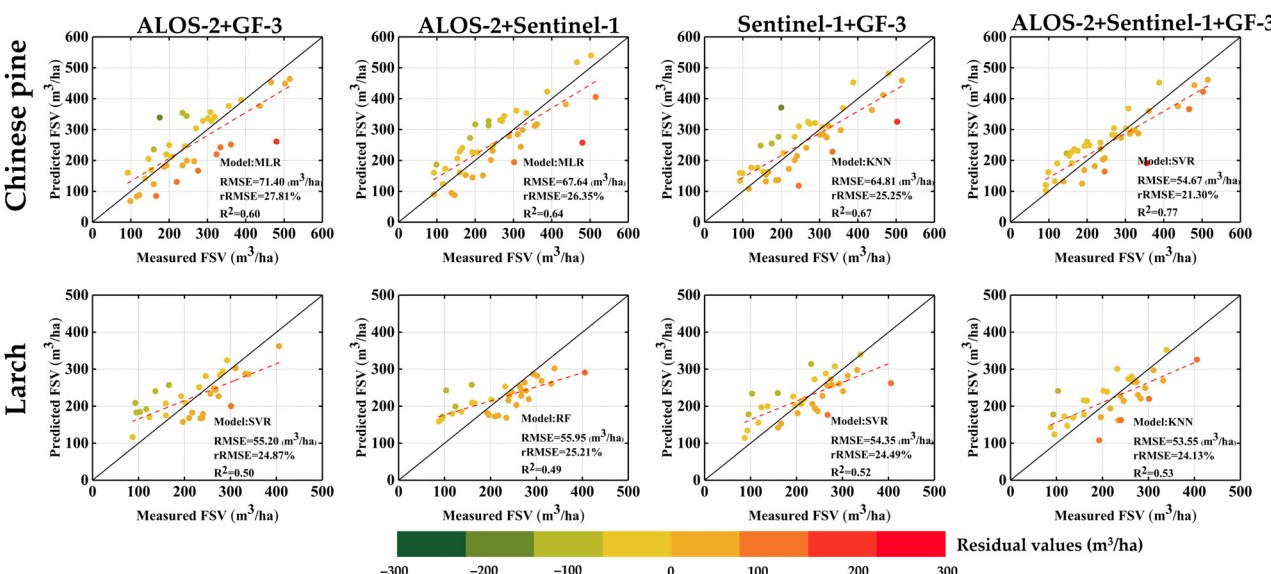

**Figure 7.** The scatter plots of estimating FSV based on combined-bands SAR data for two tree species.

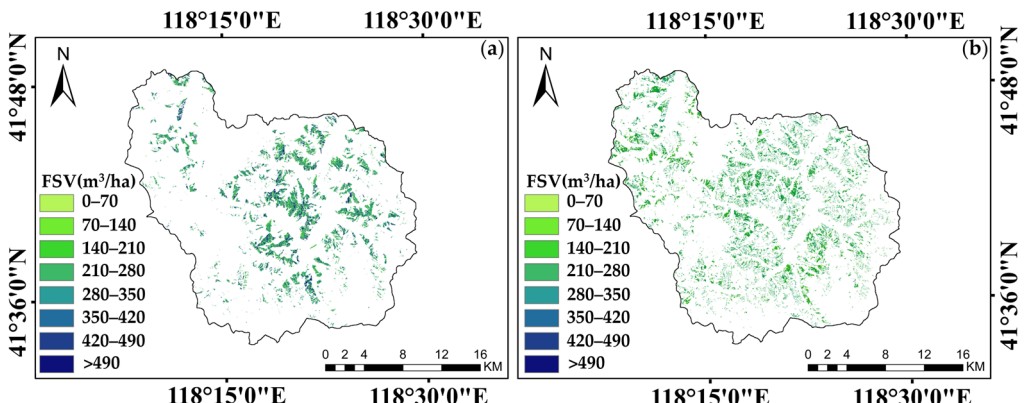

**Figure 8.** Spatial distribution of predicted FSVs obtained from ALOS-2+Sentinel-1+GF-3 in Chinese pine and Larch. (**a**) is from a machine learning model with SVR for Chinese pine; (**b**) is from a machine learning model with KNN for Larch.

## 4. Discussion

### 4.1. Polarization Response of Deciduous and Evergreen Coniferous Forests

Previous studies showed that the sensitivity between FSVs and alternative features closely relates to bands and polarization modes [35–37]. For deciduous and evergreen coniferous forests, the difference of backscattering energy in HH polarization is mainly caused by the penetration capability related to bands. Therefore, the information of trunks and branches acquired by L-band SAR images is more than that acquired by C-band SAR images [38]. Furthermore, backscattering energy in HV polarization reflected from the forest canopy is commonly lower than that in HH polarization in the same type of sensors [39]. In this study, the difference of backscattering energy in co-polarization modes highly related to bands and tree species in the non-growing season (Figure 4). The obvious gaps of backscattering energy in co-polarization modes can be observed between GF-3 and ALOS-2 in evergreen forest (Chinese pine), and the gaps were reduced in deciduous forest (Larch) in the non-growing season (Figure 4). Moreover, features extracted from Sentinel-1 with VV polarization have more potential than those extracted from ALOS-2 and GF-3 images in mapping FSV in deciduous forest in the non-growing season (Figure 5). It is inferred that polarization modes are the key factor in mapping FSV in deciduous forest without leaves [40,41].

In addition, the sensitivity between features and FSVs is influenced not only by bands and polarization modes, but also by forest structure [42–47]. Previous studies have shown that polarization features are highly correlated with forest parameters in evergreen forest [48]. Normally, backscattering energy extracted from dual SAR images with C-band mainly reflects forest canopy information, and the images with L-band reflects trunk information below the canopy [49,50]. In the non-growing season, the main difference between evergreen and deciduous forests is the depth of crowns. For deciduous forests without leaves, the effect of microwave signal penetration on the scattering energy becomes weak, and the polarization modes become the main factor affecting the scattering signal. In this study, the values of correlation between polarization features and FSV in evergreen coniferous forests (Chinese pine) were higher than those in deciduous coniferous forests (Larch) in the non-growing season. Specially, the polarization features extracted from Sentinel-1 images were highly correlated with FSV of evergreen coniferous forests (Chinese pine), but the estimated accuracy of mapped FSV using single Sentinel-1 images was the lowest among three types of sensors. For deciduous forests without leaves, the correlations between features extracted from ALOS-2 and FSV were slightly higher than those extracted from Sentinel-1, but the accuracy of mapped FSV using Sentinel-1 images was higher than that using ALOS-2 images in deciduous coniferous forests (Larch). In the non-growing season, the results implied that the accuracy of mapping FSV in evergreen coniferous forests (Chinese pine) depends on bands [28,51], while mapping FSV in deciduous coniferous forests (Larch) mainly depends on polarization modes in the non-growing season [28].

### 4.2. Combined Effects of Multi-Bands Dual Polarization SAR Images

Because of different penetration capabilities and distinct microwave scattering processes in evergreen and deciduous coniferous forests, the accuracy of mapping FSV commonly depended on the bands and polarization modes [52–54]. Previous studies have shown that the accuracy of estimating FSV using the L-band dual polarization SAR images is higher than using C-band images, caused by penetration capabilities improving the underestimation results and delaying the saturation levels [55,56]. In this study, using single-band polarization SAR images, the best accuracy of FSV estimation in evergreen coniferous forests (Chinese pine ($R^2$) ranged from 0.49 to 0.56) was significantly higher than in deciduous coniferous forests (Larch ($R^2$) ranged from 0.28 to 0.52). The number of samples with underestimated FSV in deciduous forests (Larch) was larger than that in evergreen coniferous forests (Chinese pine) (Figure 6). It is confirmed that the L- band dual polarization SAR images are more suitable for mapping FSV in evergreen forest (Chinese pine), and Sentinel-1 images with VV and VH polarization may be more suitable for mapping FSV in deciduous coniferous forests (Larch) in the non-growing season.

Previous studies have also shown that combined images with different bands and polarization modes can improve the accuracy of mapping FSV [53,57]. The degree of accuracy improvement is severely related to the combination strategies and forest structure parameters. In this study, the results demonstrated that the accuracy of mapping FSV using combined SAR images was significantly improved compared to the results derived from a single type of sensor (Figure 9). The main reason for the improvement of accuracy is that overestimation and underestimation have been greatly alleviated using combined SAR images. It is confirmed that combined effects of multi-bands dual polarization SAR images have great potential to delay saturation levels.

In addition, the accuracy of mapping FSV using combined images related to combination strategies. In this study, the results also demonstrated that there is little variation in accuracy between different combinations using various combined images with two types of sensors; the best results were obtained from a combination of three types of sensors (ALOS-2 + Sentinel-1 + GF-3). It is implied that dual polarization SAR images with various bands and various polarization modes contain different scattering information for evergreen and deciduous coniferous forests, and these complementarily make an important contribution to improving the accuracy of mapping FSV [36,54].

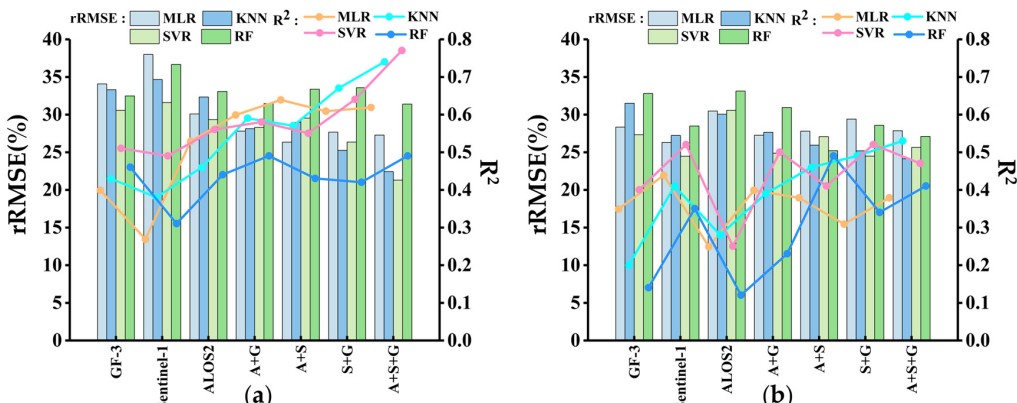

**Figure 9.** The result of mapping FSV using single band SAR images and combined SAR images. (**a**) is for Chinese pine forests and (**b**) is for Larch forests.

## 5. Conclusions

In this study, three dual polarization SAR images (GF-3, Sentinel-1, and ALOS-2) were acquired in March 2017 in planted Larch and Chinese pine forests. Then, backscattering energy of co-polarization and cross-polarization and sensitivity between FSV and all alternative features were analyzed to express the difference of bands and polarization modes in deciduous and evergreen forests in the non-growing season. The results demonstrated that the difference in backscattering energy in bands and polarization modes can be recognized in evergreen and deciduous forests in the non-growing season. Furthermore, it is also implied that polarization mode is more important than penetration capability in mapping forest FSV in deciduous forests in the non-growing season. Comparing single and dual polarization images, the accuracy of mapped FSV was obviously improved in Chinese pine (ranged from 25.60% to 29.56%) and Larch (ranged from 26.18% to 27.68%) forests using combined images with different strategies. The best results were obtained from the combination of three types of sensors (ALOS-2 + Sentinel-1 + GF-3). Additionally, it is also confirmed that compensation effects with bands and polarization modes not only have great potential to delay the saturation phenomenon, but also have the capability to reduce errors caused by overestimation. Mapping forest FSV using dual polarization SAR images is rarely conducted at non-growing season. The purpose of this study was to analyze the response of FSV to dual polarization SAR images with bands and polarization modes in the non-growing season, and to clarify the difference between bands and polarization modes in deciduous and evergreen forests. In addition, the study confirmed that the accuracy of mapping FSV using combined images related to combination strategies. To further confirm that the response of forest FSV varied with season, more dual polarization SAR images and more diverse types of forest will be employed to conduct studies for clarifying the response changes during the growing season.

**Author Contributions:** Conceptualization, H.Z. and J.L.; methodology, H.Z. and J.L.; software, P.Y., Z.L. and T.Z.; validation, H.Z., Z.L.; formal analysis, H.Z.; investigation, H.Z., Z.L., T.Z., P.Y. and T.Z.; resources, J.L. and H.L.; data curation, H.Z.; writing—original draft preparation, H.Z.; writing—review and editing, H.Z. and J.L.; visualization, H.Z.; supervision, J.L., Z.Z. and H.L.; project administration, H.L. and J.L.; funding acquisition, H.L. and J.L. All authors have read and agreed to the published version of the manuscript.

**Funding:** This research was supported by the National Natural Science Foundation of China (Project number: 32171784); the National Natural Science Foundation of China (Project number: 42030112) and the Excellent Youth Project of the Scientific Research Foundation of the Hunan Provincial Department of Education (Project number: 21B0246).



**Data Availability Statement:** The observed FSV data from the sample plots and the spatial distribution data of forest resources presented in this study are available on request from the corresponding author. Those data are not publicly available due to privacy and confidentiality reasons. The GF-3 images are available from the China Centre for Resources Satellite Data and Application website at http://www.cresda.com/CN/ (accessed on 18 March 2017); The Sentinel-1 images are available from the European Space Agency (https://www.esa.int/, accessed on 4 March 2017); Japan Aerospace Exploration Agency for the acquired ALOS-2 PALSAR-2 images (accessed on 9 March 2017) in this study.

**Conflicts of Interest:** The authors declare no conflict of interest.

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
