# Peer review of "Interpreting the Response of Forest Stock Volume with Dual Polarization SAR Images in Boreal Coniferous Planted Forest in the Non-Growing Season"

_forests, doi:10.3390/f14091700_

Round 1

Reviewer 1 Report

The task of estimating forest stock volume (FSV) using remote methods is very important, since ground measurements are extremely time-consuming and do not allow regular assessments of FSV dynamics. The paper suggests using different modes of dual polarization SAR images to evaluate FSV in the non-growing season. Measurements in the non-growing season are important for larch, increasing the penetrating power of the method, but not for pine. However, as can be seen from Table 6, there is no significant difference in the accuracy of FSV estimates between pine and larch. The average rRMSE value for the two breeds is slightly more than 30%, which, from my point of view, is not enough to use methods in assessing FSV. Nevertheless, I recommend the manuscript for publication. It seems that the measurements were carried out methodically correctly and the conducted research makes it possible to assess the prospects of the proposed method.

Author Response

We sincerely thank you for your valuable suggestions on the manuscript, which we will improve the quality of the manuscript, and in the subsequent experimental process, we will further promote our research.

Reviewer 2 Report

The paper is devoted to forest stock volume estimation using different types of SAR data and machine learning methods. The authors used images for non-growing season and compared results for deciduous (Larch) and evergreen (Chinese pine) forests. This problem is relevant, and the authors show interesting results comparing different strategies combinations. I have a few small comments to improve the quality of the manuscript.

11.       In Abstract you should give the full spelling of the abbreviation FSV.

22.       Subsection 2.5: You choose eighteen features to extract from SAR images. Where did these particular features come from, some other work (then you need references to it) or your personal experience?

33.       What software have you used for SAR image processing and machine learning? Please add a mention of it in the article.

44.       Line 155: It's probably a typo – “Concurrence Matrix” instead of “Co-occurrence matrix”.

55.       All Figures are too low resolution, so the text on them is almost unreadable.

Author Response

Thanks for your valuable suggestions and comments. We considered all your comments and made corresponding revisions. The revised manuscript has been greatly improved and we hope it can be published in the journal now. The revisions according to your comments are as follows:

Point 1:   In Abstract you should give the full spelling of the abbreviation FSV.

Response: Thanks for your suggestion. we have added the full spelling of FSV and made some red-labeled changes at the abstract of the revised manuscript.

Point 2: Subsection 2.5: You choose eighteen features to extract from SAR images. Where did these particular features come from, some other work (then you need references to it) or your personal experience?

Response: Thanks for your suggestion. We extracted these 18 features with reference to similar studies and added references in the revised manuscript.

Point 3: What software have you used for SAR image processing and machine learning? Please add a mention of it in the article.

Response: Thanks for your suggestion. We have added software about SAR image preprocessing and building machine learning models in the revised manuscript.

Point 4:  Line 155: It's probably a typo – “Concurrence Matrix” instead of “Co-occurrence matrix”.

Response: Thanks for your suggestion. It has been corrected in the revised manuscript.

Point 5: All Figures are too low resolution, so the text on them is almost unreadable.

Response: Thanks for your suggestion. We have improved the resolution of the figures in the revised manuscript.

Reviewer 3 Report

In this manuscript, the authors propose an approach for the estimation of Forest Stock Volume (FSV) using SAR images in boreal coniferous forests at the non-growing season. GF-3, Sentinel-1 and ALOS-2 sensors provide the dual polarization data that will be used as input. The manuscript is structured well, the grammar and syntax are adequate but can be improved. The authors present the materials and methods, the results and the discussion in a detailed manner. However, there are some methodological errors that are present in the work, which I believe have to be clearly addressed before the manuscript can be examined again for publication. I conclude that the manuscript requires major revisions and will be considered again only after they have been implemented.

First of all, all figures should be revised. All figures are pixelated and the information cannot be clearly read or understood.

The introduction can be expanded a bit, in order to better support the work presented and present the readers the current results of the field.

Section 2.5: I guess that the intensity values for the two distinct polarizations were used in all subsequent calculations. This is stated later in the section, after the table. Keep in mind that the readers do not necessarily have to be experts in SAR data. Please, revise that section and provide a clearer description of the data that were used. Do not leave readers guessing. Also, Table 3 is complex and cannot be clearly understood. Provide extra information and descriptions in order to support the research carried. Make the table more legible.

Furthermore, 4 machine learning regressor models are mentioned, but no information as to how they were trained, what were the hyperparameters considered/used, etc. Please provide more information.

“After getting sorted features, the sequential forward selection method and four machine learning models were employed to construct wrapped feature selection methods”. This is also unclear.

Pearson correlation is a method to detect linear correlations between two variables. So, what is proposed by the authors is to detect the linear correlation between the features used for the training and the target FSV values. Models like KNN, RF, SVM also consider nonlinear connections between the target and the features. Excluding these features based on a linear Pearson criterion may actually remove information and reduce model accuracy. This may not be the best approach for a feature selection method in this case. I would suggest a revision of this part, unless you can provide adequate literature support for this argument.

Furthermore, other methodological parts are also quite vague. The manuscript states that “Level Concurrence Matrix (GLCM) was also employed to extract textural information from each polarized intensity images”. 8 textural features are named, and various window sizes (ranged 5 to 9). If in each polarized intensity image (4 images) we apply GLCM (default distance and angle values) and extract 8 textural features per window size, for 5 different window sizes, we would get 4x8x5 = 160 texture features. Table 6 and 7 mention 4 to 22 features being used. These were also filtered using the Pearson correlation? Textures will not necessarily have a linear connection with the target FSV, so they may be marked as unimportant. In case I did not understand something correctly, please elaborate and support.

As a final remark I would suggest that the authors, after revising the manuscript, highlight the novelty of the work presented and how this may differentiate from other existing works. This was not evident in the introduction or the conclusions sections very clearly.

Grammar, syntax and vocabulary are fine. The are some minor errors that can be corrected but overall the quality of the english language is ok. 

Author Response

Point 1: all figures should be revised. All figures are pixelated and the information cannot be clearly read or understood.

Response: Thanks for your suggestion. We have improved the resolution of the pictures in the manuscript.

Point 2: The introduction can be expanded a bit, in order to better support the work presented and present the readers the current results of the field.

Response: Thanks for your suggestion. We have expanded the introduction section appropriately.

Point 3: Section 2.5: I guess that the intensity values for the two distinct polarizations were used in all subsequent calculations. This is stated later in the section, after the table. Keep in mind that the readers do not necessarily have to be experts in SAR data. Please, revise that section and provide a clearer description of the data that were used. Do not leave readers guessing. Also, Table 3 is complex and cannot be clearly understood. Provide extra information and descriptions in order to support the research carried. Make the table more legible.

Response: Thanks for your suggestion. We have revised it and added additional information to supplement it.

Point 4: Furthermore, 4 machine learning regressor models are mentioned, but no information as to how they were trained, what were the hyperparameters considered/used, etc. Please provide more information.

Response:Thanks for your suggestion. All four regression models were constructed in R software, where the parameters to be tuned for KNN are the number of neighbours K, the parameters to be tuned for RF are the number of random trees (ntree) and the number of randomly chosen variables to split each node in the tree (mtry), the parameters to be tuned for SVM are the penalty coefficient (C) and gamma, and no parameters are needed to be tuned for MLR.

Point 5: “After getting sorted features, the sequential forward selection method and four machine learning models were employed to construct wrapped feature selection methods”. This is also unclear.

Response: Thanks for your suggestion. This refers to the features sorted from high to low based on Pearson correlation. Then, the features are gradually added to the model using the sequential feature selection method, while the features that contribute poorly to the model accuracy are removed until the estimated model reaches the best accuracy. Finally, the optimal feature set is created.

Point 6: Pearson correlation is a method to detect linear correlations between two variables. So, what is proposed by the authors is to detect the linear correlation between the features used for the training and the target FSV values. Models like KNN, RF, SVM also consider nonlinear connections between the target and the features. Excluding these features based on a linear Pearson criterion may actually remove information and reduce model accuracy. This may not be the best approach for a feature selection method in this case. I would suggest a revision of this part, unless you can provide adequate literature support for this argument.

Response: Thanks for your suggestion. Comment 6 was responded to together with comment 7.

Point 7: Furthermore, other methodological parts are also quite vague. The manuscript states that “Level Concurrence Matrix (GLCM) was also employed to extract textural information from each polarized intensity images”. 8 textural features are named, and various window sizes (ranged 5 to 9). If in each polarized intensity image (4 images) we apply GLCM (default distance and angle values) and extract 8 textural features per window size, for 5 different window sizes, we would get 4x8x5 = 160 texture features. Table 6 and 7 mention 4 to 22 features being used. These were also filtered using the Pearson correlation? Textures will not necessarily have a linear connection with the target FSV, so they may be marked as unimportant. In case I did not understand something correctly, please elaborate and support.

Response: Thanks for your suggestion. The Pearson correlation coefficients used in the manuscript is a commonly used method of feature selection, which may not be optimal but has been shown to be effective. Although there are some differences with the use of other rankings such as Random Forest Importance Ordering, DC Ordering, etc., we have added relevant references in the revised manuscript. The topic of our article focuses on the FSV estimation difference of bands and polarization modes in the deciduous and evergreen forest during the non-growing season and clarifies the effects of bands, polarization modes, and tree species on the FSV estimation. The conclusions of our manuscript will not be affected by the feature selection method used. Then, 6 polarization intensity images obtained from three SAR images, GF-3, Sentinel-1, and ALOS2. By applying GLCM to extract 8 texture features respectively, 6×8=48 texture features will be obtained. Three windows of 5×5, 7×7, and 9×9 were applied to obtain a total of 48×3=144 texture features. I have modified it in the manuscript and the changes are given by adding red markers.

Point 8: As a final remark I would suggest that the authors, after revising the manuscript, highlight the novelty of the work presented and how this may differentiate from other existing works. This was not evident in the introduction or the conclusions sections very clearly.

Response: Thanks for your suggestion. We have added the conclusion and abstract sections to emphasize the highlight. The highlight of this manuscript is that mapping forest FSV using dual-polarization SAR images is rarely conducted at the non-growing season. The purpose of this study is to analyze the response of FSV to dual-polarization SAR images with bands and polarization modes at non-growing season, and to clarify the difference of bands and polarization modes in deciduous and evergreen forests. In addition, the study has confirmed that the accuracy of mapping FSV using combined images is related to combination strategies.

Round 2

Reviewer 3 Report

The authors have addressed most of the comments. The manuscript can now be accepted for publication. There are a few points that need to be address before that though.

The images still appear blurry and pixelated to me, and no information can be clearly read. I would advise the authors to be extremely cautious on the final manuscript version and incorporate high quality figures as requested in the previous revision. 

I would suggest that the authors highlight what were the most important features selected by the sequential feed forward feature selection method and if there are some patterns observed (e.g some features being selected most of the time, etc..).

Finally, the answer to point 4 should be added to the manuscript. 

After reading the final manuscript, I would suggest that the authors perform some editing of the manuscript to improve English language. As it is, the manuscript is a little difficult to understand and to follow easily in certain points. Overall though, grammar and syntax are adequate.

Author Response

Point 1: The images still appear blurry and pixelated to me, and no information can be clearly read. I would advise the authors to be extremely cautious on the final manuscript version and incorporate high quality figures as requested in the previous revision. 

Response 1: Thanks for your suggestion. We have improved the resolution of the figures in the revised manuscript.

Point 2: I would suggest that the authors highlight what were the most important features selected by the sequential feed forward feature selection method and if there are some patterns observed (e.g some features being selected most of the time, etc..).

Response 2: Thanks for your suggestion. The optimal features determined from different data sets are different, so it is difficult for us to give a particular optimal feature that is applicable to all remote sensing data sets. However, texture features are always determined to be optimal in each data set, which we have already mentioned in the manuscript.

Point 3: Finally, the answer to point 4 should be added to the manuscript. 

Response 3: Thanks for your suggestion. In the revised manuscript,We have added the answer to point 4.

Point 4: After reading the final manuscript, I would suggest that the authors perform some editing of the manuscript to improve English language. As it is, the manuscript is a little difficult to understand and to follow easily in certain points. Overall though, grammar and syntax are adequate.

Response 4: Thanks for your suggestion. We have performed some editing of the manuscript to improve the English language.